# Machine Learning Techniques for Evaluating Concrete Strength with Waste Marble Powder

**DOI:** 10.3390/ma15175811

**Published:** 2022-08-23

**Authors:** Nitisha Sharma, Mohindra Singh Thakur, Parveen Sihag, Mohammad Abdul Malik, Raj Kumar, Mohamed Abbas, Chanduveetil Ahamed Saleel

**Affiliations:** 1Department of Civil Engineering, Shoolini University, Solan 173229, Himachal Pradesh, India; 2Department of Civil Engineering, Chandigarh University, Mohali 140413, Punjab, India; 3Engineering Management Department, College of Engineering, Prince Sultan University, Riyadh 11586, Saudi Arabia; 4Faculty of Engineering and Technology, Shoolini University, Solan 173229, Himachal Pradesh, India; 5Electrical Engineering Department, College of Engineering, King Khalid University, P.O. Box 394, Abha 61421, Saudi Arabia; 6Computers and Communications Department, College of Engineering, Delta University for Science and Technology, Gamasa 35712, Egypt; 7Department of Mechanical Engineering, College of Engineering, King Khalid University, P.O. Box 394, Abha 61421, Saudi Arabia

**Keywords:** concrete, compressive strength, flexural strength, support vector machines, gaussian processes, linear regression

## Abstract

The purpose of the research is to predict the compressive and flexural strengths of the concrete mix by using waste marble powder as a partial replacement of cement and sand, based on the experimental data that was acquired from the laboratory tests. In order to accomplish the goal, the models of Support vector machines, Support vector machines with bagging and Stochastic, Linear regression, and Gaussian processes were applied to the experimental data for predicting the compressive and flexural strength of concrete. The effectiveness of models was also evaluated by using statistical criteria. Therefore, it can be inferred that the gaussian process and support vector machine methods can be used to predict the respective outputs, i.e., flexural and compressive strength. The Gaussian process and Support vector machines Stochastic predicts better outcomes for flexural and compressive strength because it has a higher coefficient of correlation (0.8235 and 0.9462), lower mean absolute and root mean squared error values as (2.2808 and 1.8104) and (2.8527 and 2.3430), respectively. Results suggest that all applied techniques are reliable for predicting the compressive and flexural strength of concrete and are able to reduce the experimental work time. In comparison to input factors for this data set, the number of curing days followed by the CA, C, FA, w, and MP is essential in predicting the flexural and compressive strength of a concrete mix for this data set.

## 1. Introduction

The emission of CO_2_ is a major environmental concern. It is important to take specific steps in order to minimize the amount of CO_2_ emissions into the environment. India ranks third in the world in terms of CO_2_ emissions. Concrete is a major contributor to increased CO_2_ levels in the atmosphere. Concrete consists of three major ingredients such as cement, which contributes to the improvement of concrete’s strength; aggregates, which provides the bulkiness to the concrete; and water, which reacts with cement, which leads to the hydration process and provides strength to concrete. Except for cement, all of the other components are readily available across the world. The only method to produce cement is to manufacture it. CO_2_ is released during cement production, which pollutes the environment. Over-exploitation of materials is on the rise as a result of their widespread use.

The most efficient method for reducing the negative effects that concrete has on the environment is to reduce the amount of cement that is used in its production. This goal can be accomplished through the use of alternative building materials [1,2,3]. Concrete is frequently utilized in industries, which increases its demand and will result in its depletion. It would be economically viable to employ waste material in infrastructure development if it met the required specifications [4,5]. Various hazardous wastes are generated as a result of industrialization, which can be managed by incorporating them into basic concrete components. Some mineral additions in concrete, for example, fly ash, silica fumes, and blast furnace slag, might increase the demand for water in the concrete mix, which can be addressed with the use of a superplasticizer [6].

The bond strength of the concrete matrix can be improved by using calcium carbonate whiskers and basalt fiber in the concrete matrix [7]. Furthermore, silica fume can be used in place of cement in concentrations ranging from 0 to 10%, which improves the interface bond. Apart from silica-fume and basalt fiber, some researchers use coconut fiber as a partial replacement for cement [8]. As an alternative [9,10,11,12], WMP can be used instead of cement or fine aggregates. WMP is a promising resource that may be used to partially replace cement and sand. CaO, SiO_2_, Al2O3, and Fe_2_O_3_ are significant constituents, whereas MgO, SO_3_, K_2_O, and Na_2_O are minor constituents. India is the world’s third largest producer of marble trash. From the beginning of the mining process until the end, waste is produced. If the waste created by the marble business is not correctly utilized, it may have an adverse effect on both the environment and the economy. Marble is frequently used for ornamental purposes, increasing its market demand. Marble manufacturing results in a variety of chemical formations that are classified as hazardous waste. Waste disposal is not cost-effective, and the environment is also a concern. The use of properly integrated industrial waste can help to minimize the amount of cement required in the concrete mix [13]. Marble sludge may be recycled and used as a major component of concrete mix, which can then be used as a construction material or in road pavements, among other things [14]. In comparison to the nominal mix, Soliman [15] found that as the amount of WMP in the nominal concrete in place of cement increases, it deteriorates concrete strength. The effect of partially replacing cement with WMP and finding suggests that tensile strength with WMP begins to deteriorate as the amount of waste marble powder increases [16]. Replacement of 10% WMP with cement improves the tensile behavior of concrete [17]. Many researchers use waste marble powder as a replacement for cement by weight [18,19]. When an experiment was carried out by Kelestemur [20] on a concrete mix containing various amounts of glass fiber, it can be seen that by adding WMP, it achieves the highest compressive strength. The capillarity properties of SCC were altered by adding WMP in place of cement, and it was found that the addition of WMP had no effect on workability, but tensile strength decreased [21]. Ashish [22] investigated the possibility of using WMP as a partial replacement for fine aggregate and cement amalgam. Uysal and Yilmaz [23] performed experiments incorporating lime, basalt and WMP as Portland cement substitutes. Hebhoub [24] found that when gravel or both were substituted with sand, the 28-day tensile strength was higher. The use of marble powder as a filler was found to be adequate by Corinaldesi [25]. As per literature, use of 10% WMP in lieu of sand during the 28 days of curing offers maximum compressive strength at roughly the same workability as cement replacement. Due to its fineness, marble powder offers good cohesiveness to mortar. SCC was used as an alternative to WMP [26]. At the 28-day curing time, Demirel [27] discovered that when the fine particles in the matrix were replaced with waste marble powder, the porosity decreased but the compressive strength increased. The use of WMP as a filler affected the concrete’s mechanical and physical properties [28]. Talah [29] looked at the results of tests carried out with and without WMP and found that MP can be used to make high-performance concrete.

Soft computing approaches are now being used by researchers to solve the issues [30,31,32,33,34,35,36,37,38]. The hardened property of concrete is primarily determined by the amount of cement, aggregates, water, admixtures, and wastes used in the mix. These factors can be used as an input parameter to aid in the prediction of the final result. The linear and non-linear regression approaches are mostly used in traditional methods for anticipating outcomes. However, in recent years, AI approaches such as ANN, LR, GMDH, RF, SVM and RT have been used to estimate the mechanical characteristics of concrete [39,40]. The majority of the study is focused on predicting the mechanical characteristics of concrete mixes. Several regressions, NNT, and ANFIS models are constructed by Sobhani [32] using concrete components as input parameters in the estimation of the hardened property of concrete. The outcome demonstrates that the NNT and ANFIs algorithms are more accurate compared to proposed standard regression algorithms for predicting the 28-day compressive strength. Madandoust [41] used GMDH, NNT, and ANFIs modelling for predicting the CS of concrete using cementitious materials. A genetic algorithm was used in the study to build the GMDH kind of neural network. In order to forecast output strength, parameters such as length to diameter ratio, core diameter, and so on were used as input. The study was directed by Ayat [42] to evaluate the affectability of the constructed model to some basic factors influencing concrete compressive strength. It was discovered that the suggested ANNs model’s performance was outstanding with a highly effective tool for the simulation of the compressive strength with LF. Deepa [43] estimates the CS of concrete utilizing classification methods such as M5P Tree models, Multilayer Perceptron’s, and Linear Regression. The results show that tree-based algorithms predict better results for concrete strength.

The purpose of this study was to determine whether or not machine learning could be used to make accurate predictions regarding the compressive and flexural strengths of concrete mixtures by adding waste marble powder in place of cement and sand together. To achieve the objective of the study, experimental data obtained by using waste marble powder as a partial replacement with cement and sand in the following proportions: 0–100, 25–75, 50–50, 75–25, and 100–0, respectively, was analyzed with soft computing techniques such as Support Vector Machines (SVM), Gaussian Processes (GP), and Linear Regressions (LR). The findings were than compared by using statistical parameters such as Coefficient of correlation (CC), Mean absolute error (MAE), Root mean square error (RMSE), relative absolute error (RAE), and Root relative squared error (RRSE). Further to identify the most relevant parameters (C, FA, CA, w, MP, and CD) in terms of predicting the compressive and flexural strength of concrete mixes and to ascertain the best modelling technique for predicting the compressive and flexural strength of concrete, sensitivity analysis has been carried out.

## 2. Machine Learning Techniques

### 2.1. Support Vector Machine (SVM)

SVM, is a complex machine learning algorithm which is used by researchers to solve challenging engineering problems such as categorization, forecasting, and regression analysis. It is a part of artificial intelligence [44,45]. The SVM analysis approach includes training and testing sub-sets coupled with input and output. The optimal margin classifier is one of two methods used in SVM analysis to separate the linear classifier. Another alternative is to use the kernel function technique. The fixed mapping technique is used after mapping the input data with n-dimensional characteristics. There has been no change in the actual input space when the kernel mapping is applied to actual observations during a high-dimensional feature [45].

#### 2.1.1. Bagging

Bagging is used to improve poor learning machine predictions [46]. Bagging factor determines how much of the original database will be recombined [47]. Each model uses bootstrap resampled observed data. Bagging algorithms have three steps: Bootstrapping uses replacements to create a new training set. Distributed voting is used to associate classifier results. This approach improves generalisation and classification variance. To use this method, the basic classifier must be imbalanced; otherwise, no classification will arise. The K-Nearest Neighbor classifier is stable, while the MLPC is not [48]. In this study, bagging is applied to check the reliability of the SVM model for predicting the strength property of concrete. 

#### 2.1.2. Stochastic

Stochastic meta-assembly improved the traditional techniques-based model. It is used to solve nonlinear engineering problems. In some studies, stochastic methods improve random forest models [31]. The designing process is filled with uncertainties, such as natural randomness of physical quantities, Model assumptions due to shortcomings in the computer model compared to real structural data, and statistical assumptions during the identification of a quantity due to a lack of data [49]. A stochastic process consists of arbitrary variables whose values are determined by Q’s state vector. Indexing the gathering uses another set Z.

### 2.2. Gaussian Processes (GPs)

During the preceding years, a substantial amount of work has been carried out in subject of machine learning. GP is a machine learning method for analyzing the models’ using kernels which provides a hands-on approach to learning about kernel [50]. It is a set of random variables with a mutual normal distribution for each discrete variable. The two fundamental functions used in the GP l(a) are the mean function m(a) and the kernel function n(a, a′). It states that:l(a) = GP(m(a), n(a,a′)),(1)

The GP’s primary goal is to find, how the input variables can be used to accomplish the objective. For every objective value, such as y, there is an arbitrary regression function l(a) and independent Gaussian noise (ϧ), i.e.,
y = l(a) + ϧ(2)

Gaussian noise with a mean of zero and variance of (σ_n_)^2^, i.e., ϧ~L(0, σ_n_^2^). Then Equation (1) develops Equation (3):l(a) = GP(m(a), n(a,a′) + σ_n_^2^I),(3)

The identity matrix is represented by I.

### 2.3. Linear Regression (LR)

Linear regression is a regression analysis model that employs linear equations to represent the relationship of the two or more dependent and independent variables [32]. The primary goal of a linear regression model is to identify the linear line that best predicts the relationship of the dependent and independent variables. It is only conceivable if the total of the squares of the vertical lines is less than the specified line. Linear regression models are made up of dependent and independent variables, such as d and z from the given dataset. To discover the optimum numerical forecast, it forms a simple mathematical equation. It also aids in determining the correlation coefficient capable of describing the variances in the dataset. The closer the value is to one, the more dependable the data. [43].
d = c_0_ + c_1_z_1_ + c_2_z_2_ + C_3_z_3_ + C_4_z_4_ + C_5_z_5_ + C_6_z_6_ +…(4)

#### Objective of the Study

A detailed review of the literature revealed that no study had employed these modelling techniques to predict the compressive and flexural strength of concrete mix. Considering their application in construction material, an attempt was made to evaluate their potential in predicting the compressive and flexural strength of concrete mixtures by adding waste marble powder in place of cement and sand together using laboratory data.

## 3. Methodology

The following methods must be followed in order to achieve the study’s purpose of estimating concrete compressive and flexural strength.
Experimental data on the compressive and flexural strength.Used soft computing techniques.

### 3.1. Collection of Data

For the prediction of the compressive and flexural strength, sufficient data was necessary, which was supplied by performing experimental research using waste marble powder in various proportion as a partial replacement of cement and sand.

### 3.2. Experimental Investigation

Concrete with complicated structures is comprised of multiple complex materials, making them difficult to comprehend, e.g., the problem of matrix/aggregate interactions, the region between the aggregate and the matrix, the size of aggregates, etc. The size and quantity of particles, for instance, have been found to affect the fracture energy and fracture toughness of concrete. The matrix can be composed of several materials in variable amounts, which can alter their qualities, such as the w/C and C/S, among others. The same holds true for aggregate, which may be composed of pebbles, with smooth, rounded surfaces, or crushed rock with irregular shapes, varying chemical and mineralogical compositions, and extremely rough surfaces. The aggregates appearance and texture have a big effect on how well they fit together with mortar. Further, due to the large specific surface area of the marble powder (MP), which is obtained from the processing of marble waste sludge, it has the potential to be utilized as a filler in concrete. The following materials and testing methods were used on 189 cubes and 188 beam specimens with dimensions of 150 × 150 × 150 mm and 700 × 150 × 150 mm, respectively.

#### 3.2.1. Aggregate

Coarse aggregate with nominal sizes of 10 and 20 mm was used in the concrete mix. The aggregate’s particle size distribution was graded [51]. According to ASTM C-128 and ASTM C-127 [52,53], specific gravity, crushing, and impact were found to be 2.61%, 23.67%, and 6.74%, respectively. Table 1 shows the characteristics of fine and coarse aggregate.

#### 3.2.2. Cement

As per ASTM C150 [54] Type-I cement was used in this study. ASTM C184, ASTM C187, ASTM C151, ASTM C188, ASTM C191 [55,56,57,58,59] was used to calculate the mechanical properties of cement. Values obtained for fineness, Consistency, Soundness, and Specific Gravity were 5.77, 32, 3.33, and 3.1 and initial and final setting time of cement was 40 and 360 min.

#### 3.2.3. Marble Powder

A nearby marble market supplied the WMP. The physical properties of WMP obtained for fineness, and Specific Gravity were 2.01% and 2.44 g/cm^3^.

#### 3.2.4. Mix Design

Batches were created according to the prescribed quantities of cement, sand and marble powder, while other elements such as coarse aggregates, and water/cement ratio were utilized in the same amounts throughout. Different sets of specimens were made, each consisting of three cubes and three beams. A total of 189 cubes and 188 beams were created. The proportion of reference specimens, as well as mixes with 5, 10, 15, and 20% replacement, is listed in Table 2. In terms of compressive and flexural strength, the mechanical properties of marble powder (MP) modified concrete were investigated as a partial replacement of cement and sand in proportions of 0–100, 25–75, 50–50, 75–25, and 100–0% where 5% of MP, designated as MP 5(C0:S100), MP 5(C25:S75), MP 5(C50:S50), MP 5(C75:S25), MP 5(C100:S0), 10% of MP, designated as MP 10 (C0:S100), MP 10(C25:S75), MP 10(C50:S50), MP 10(C75:S25), MP 10(C100:S0), 15% of MP designated as MP15 (C0:S100), MP 15(C25:S75), MP 15(C50:S50), MP 15(C75:S25), MP 15(C100:S0),and 20% of MP, designated as MP 20(C0:S100), MP 20(C25:S75), MP 20(C50:S50), MP 20(C75:S25), and MP 20 (C100:S0), with cement and sand in a percentage of 0–100, 25–75, 50–50, 75–25 and 100–0, respectively.

Additionally, modified concrete was mixed in a mixer, and the necessary substitution for ordinary cement was carried out during the mixing process. The mixes were mixed to with MP, and (150 × 150 × 150) mm cubes and (150 × 150 × 700) mm beams were created. After that, samples were given a certain number of days in curing tanks to cure (CD). The CD that was chosen for this investigation ranged from 7 to 180 days. After curing, cube and beam samples were sun-dried and tested in the institute’s lab using a standard compression and flexural testing machine. The CS and FS tests were carefully performed to determine the mix proportion from the collection of casted samples suitable to achieve approximately the same or better compressive and flexural strength for a selection of percentage substitutions [60,61].

#### 3.2.5. Data Collection

The dataset is critical for predicting the outcome. The observations collected from laboratory study were divided into two sub-sets at random, with 70–30 ratio for the training and testing sub-sets, respectively.

In this paper, three techniques were used, namely LR, SVM, SVM Bagging and SVM Stochastic and GP, with input parameters such as C, FA, CA, w, MP, and CD, to achieve the desired outcome, with CS and FS as the output parameter, using Weka 3.9. Table 3 and Table 4 lists the characteristics of the total, training, and testing datasets for CS and FS, respectively, and user defined parameters listed in Table 5. Each model’s performance was evaluated using CC, MAE, RMSE, RAE, and RRSE. These variables were beneficial in finding the best model. A greater CC value and a lower error value suggest better results. A significant number of studies resulted in these user-defined ideal settings for diverse techniques. The best settings dictated the performance of each model. The optimal settings must be carefully determined because they will have an impact on the model’s performance. As a result, the parameters in these examples were ideal for both training and testing datasets.

## 4. Evaluation Parameters

Evaluating parameters must be applied to the applicable algorithms in order to verify their performance. Coefficient of correlation (CC), Mean absolute error (MAE), Root mean square error (RMSE), relative absolute error (RAE), and Root relative squared error (RRSE) were utilized as evaluating parameters in this study.

Coefficient of correlation (CC): The coefficient of correlation (R) was utilised as the primary criterion to assess the effectiveness of the created models. The following equation was used to acquire the value of CC:(5)$$\mathrm{CC}=\frac{y\left(\sum_{i=1}^{y}\mathrm{ZW}\right)-\left(\sum_{i=1}^{y}Z)(\sum_{i=1}^{y}W\right)}{\sqrt{[y\sum_{i=1}^{y}Z^{2}-{\left(\sum_{i=1}^{y}Z\right)}^{2}]\;}\sqrt{[y\sum_{i=1}^{y}W^{2}-{\left(\sum_{i=1}^{y}W\right)}^{2}]\;}}$$

Root mean square error (RMSE): Mean Squared Error is the most often used method for determining a model’s success. It is the square root of Mean-Squared-Error with the same dimensions as the estimated values. The root mean square error (RMSE) is computed as follows:(6)$$\mathrm{RMSE}=\surd \frac{1}{y}\left(\sum_{i=1}^{y}{\left(W-Z\right)}^{2}\right)$$

Mean absolute error (MAE): The mean absolute error is used to assess the accuracy of numerical estimation. The mean absolute error (MAE) is calculated as follows:(7)$$\mathrm{MAE}=\frac{1}{y}\left(\sum_{i=1}^{y}\left|W-Z\right|\right)$$

Relative absolute error (RAE): The relative absolute error (RAE) is a metric used to assess the performance of a predictive model. It is calculated by using following equation:(8)$$\mathrm{RAE}=\frac{\sum_{i=1}^{y}\left|Z-W\right|}{\sum_{i=1}^{y}\left(\left|Z-\bar{Z}\right|\right)}$$

Root relative squared error (RRSE): It is defined as the square root of a predictive model’s sum of squared errors normalized by the sum of squared errors of a simple model. To put it another way, it is the square root of the Relative Squared Error (RSE) and calculated by using formula:(9)$$\mathrm{RRSE}=\sqrt{\;\frac{\sum_{i=1}^{y}{\left(Z-W\right)}^{2}}{\sum_{i=1}^{y}{\left(\left|W-\bar{W}\right|\right)}^{2}}}$$Z = Observed readings; $\bar{Z}$ = Average of observed readings; W = Predicted readings; y = Number of readings.

Coefficient of correlation (CC) values vary from −1 to +1; the greater the CC number, the better the anticipated outcomes. Lower the values better will be the outcomes, i.e., if computed error is low, it predicts better output results [62].

## 5. Result and Discussion

To solve complex engineering problems, researchers apply a variety of machine learning techniques [63,64,65,66,67,68,69,70,71]. This study examined SVM, LR, and GP-based models, and the findings from this analysis are discussed in this section.

### 5.1. SVM Based Assessment

PUK base function is used in this model, along with certain user-defined parameters such as C and omega (O), and sigma (S) A large number of trials were conducted in order to arrive at the best result, which was the greatest CC value with the least errors. The dataset utilized in this study produced the best results with a c value of 1.5 and O = 4, S = 1. Table 6 lists the performance metrics of SVM for both the training and testing datasets. For the training and testing phases, the CC values were 0.9110 and 0.8153, the RMSE values were 2.2287 and 3.0367, the MAE values were 1.5123 and 2.3456, the RAE values were 33.28% and 67.87%, and the RRSE values were 41.79% and 74.13%, respectively. The agreement plot between actual and predicted flexural strength of concrete mix is shown in Figure 1.

The radial basis function kernel (RBF kernel) is used in this model, along with certain user-defined parameters such as C and gamma (Ɣ). A large number of trials were conducted in order to arrive at the best result for compressive strength of concrete, which was the greatest CC value with the least errors. The dataset utilized in this study produced the best results with a c value of 5 and a Ɣ value of 10. Table 7 lists the performance metrics of SVM for both the training and testing datasets. For the training and testing phases, the CC values were 0.9545 and 0.945, the RMSE values were 2.3289 and 2.2908, the MAE values were 1.2472 and 1.7305, the RAE values were 18.33% and 28.46%, and the RRSE values were 30.04% and 33.93%, respectively. The agreement plot between actual and predicted compressive strength of concrete mix is shown in Figure 2.

Table 6 and Table 7 presents analytical performance measures for training and testing datasets to aid in the evaluation of proposed models for FS and CS of concrete, respectively. To achieve a precise performance, the CC should be close to one and the RMSE and MAE values should be close to zero. The SVM algorithm predicts compressive strength better than flexural strength with higher CC (0.9450) value.

Table 6 and Table 7 show that when SVM, SVM-Bagging, and SVM-Stochastic were evaluated, SVM-Stochastic is more reliable for predicting compressive strength with highest CC value of 0.9462.

### 5.2. GPs Based Assessment

Gaussian Processes are a type of regression that uses a PUK function and certain user-defined parameters such as L, omega (O) and sigma (S). Several experiments were conducted in order to arrive at the optimal value, which was the maximum CC value with the lowest errors. The dataset utilized in this investigation yielded the best results with L values of 0.01, O = 3 and S as 3. Table 6 lists the performance metrics for both the training and testing datasets for general practice. The CC values for the training and testing phases were 0.9170 and 0.8235, respectively, with RMSE values of 2.1276 and 2.8527, MAE values of 1.7012 and 2.2808, RAE values of 37.44% and 66%, and RRSE values of 39.90% and 69.63%. Figure 3 depicts the agreement plot between the actual and predicted flexural strength of concrete mix.

The dataset utilized in this investigation yielded the best results with L values of 0.1 and O as 0.1 and S as 1.0. Table 7 lists the performance metrics for both the training and testing datasets for general practice. The CC values for the training and testing phases were 0.9704 and 0.9429, respectively, with RMSE values of 1.8734 and 2.4377, MAE values of 1.28 and 1.8728, RAE values of 18.81% and 30.80%, and RRSE values of 24.16% and 36.10%. Figure 4 depicts the agreement plot between the actual and predicted compressive strength of concrete mix.

### 5.3. LR Based Assessment

LR based model predict the relationship between two variables or factors. The results of Table 6 show the linear regression model technique for flexural strength for both training and testing dataset Figure 5 is the graphical representation of actual versus predicted flexural strength with coefficient of correlation CC is 0.7802, RMSE is 2.7717, MAE is 2.3822, RAE is 68.93% and RRSE is 67.66%, for testing dataset as listed in Table 6. Equation (10) was developed by LR algorithm from Weka 3.9.5 for the prediction of flexural strength of concrete.
(10)FS=0.0563×curing days +4.4332

LR based model predict the relationship between two variables or factors. The results of Table 7 show the linear regression model technique for the compressive strength for both training and testing dataset. Figure 6 is the graphical representation of actual versus predicted compressive strength with coefficient of correlation CC is 0.8712, RMSE is 3.3921, MAE is 2.6909, RAE is 44.26% and RRSE is 50.24%, for testing dataset listed in Table 7. Equation (11) was developed by LR modelling through Weka 3.9.5 for the prediction of compressive strength for a given set of data.
(11)$$\mathrm{CS}=-0.0164\;*\;\mathrm{Marble}\;\mathrm{Powder}\left(\mathrm{kg}/m^{3}\right)+0.087\;*\;\mathrm{Curing}\;\mathrm{Days}\;+13.6919$$

## 6. Comparison

Various machine learning approaches were used in this research. When these models are compared, it appears that the GP model outperforms the others for both training and testing datasets for FS. Figure 7 represents the scatter plot for the observed and predicted FS values using SVM, GP, and LR. Table 6 indicate that the GP model has the greatest CC value for both training and testing datasets for FS, i.e., 0.9170 and 0.8235, respectively, as well as the lowest error values, i.e., MAE (2.2808) and RAE (66%), for the testing dataset. It can be seen from Figure 8 that the error band width is lesser in GP based model as compared to other applied models for the prediction of FS of concrete mix [63]. Figure 1, Figure 3 and Figure 5 for the SVM, GP, and LR testing models show that the GP model has a high correlation with the maximum R^2^ value for the flexural strength of concrete, whereas Figure 2, Figure 4 and Figure 6 for the SVM, GP, and LR testing models show that the SVM-Stochastic model has a high correlation for the compressive strength of concrete.

Figure 9 represents the scatter plot for the observed and predicted compressive strength values using SVM, GP, and LR. Table 7 indicate that the SVM model has the greatest CC value for testing datasets for the compressive strength, i.e., 0.9450, as well as the lowest error values, i.e., MAE (1.7305), RMSE (2.2908), RAE (28.46%) and RRSE (33.93%) for the testing dataset. It can be seen from Figure 10 that the error band width is lesser in SVM based model as compared to other applied model for the prediction of CS of concrete mix. Figure 11 represents the scatter plot between no. of observation and ratio of actual and predicted strength. It shows that SVM-Schotastic based model is reliable for compressive strength prediction followed by SVM [64] as compared to GP model for flexural strength because the values are more scatter in the ratio of actual and predicted flexural strength.

In addition to the actual value, quartile values of 25%, 50%, and 75% were assessed for the evaluation of FS and CS of concrete containing marble powder, as shown in Table 8. The inter quartile range (IQR) of SVM-Schotastic and SVM Bagging is closer to the IQR of real data, as seen in Figure 12 and Figure 13 for FS and CS, respectively.

## 7. Sensitivity Analysis

Using sensitivity analysis, the most significant factor among input factors for predicting the CS and FS with WMP was determined. Since GP and SVM-Schotastic model performed the best among the other models for FS and CS for this dataset, sensitivity analysis was carried out on it by changing the input combination and taking out one input parameter at a time, as shown in Table 9 and Table 10. Statistical assessment metrics such as CC, MAE, and RMSE were used to assess each model’s performance [72,73,74,75,76,77,78,79,80,81,82]. Table 9 and Table 10, demonstrates that the number of curing days followed by CA, C, w and MP is critical in predicting the flexural and compressive strength of a concrete mix. Due to the pozzolanic reactions, concrete recovers 60% of its strength after 7 days of curing and increases by 99% after 28 days, resulting in a low CC value after removing the aforementioned characteristic [76,77,78,79]. The pozzolanic reaction is a slow process, and as the curing period lengthens, the amount of gel produced in the mix increases, resulting in greater strength [65]. Hydration products form when water is added to cement mixes, filling gaps between aggregate particles and increasing the density and strength of the concrete.

## 8. Conclusions

This study compared the CS and FS using WMP by three machine learning techniques: SVM, LR, and GP based models. CC, MAE, RMSE, RAE, and RRSE were applied to evaluate the performance of these models. The following is a summary of the study’s findings:

The GP and SVM model’s results were shown to be the best for predicting concrete flexural and compressive strength of concrete mix, respectively. For the testing dataset, GP predicts better outcomes followed by SVM with CC values of 0.8235 and 0.8153, lower MAE values of 2.2808 and 2.3456, and lower RMSE values of 2.8527 and 3.0367, respectively, for flexural strength of concrete. SVM-Schotastic predicts better results with CC values of 0.9462, lower MAE values of 1.8104 and lower RMSE values of 2.3430 for compressive strength of concrete. The scatter plot reveals that the GP and SVM-Schotastic has the smallest error band width and is a strong match for predicting flexural and compressive strength as output, respectively.

In comparison to input factors for this data set, the number of curing days followed by the CA, C, w and MP is essential in predicting the flexural and compressive strength of a concrete mix for this data set. These findings are restricted to a 0–20% cement sand substitute with marble powder. Additionally, CD is a more sensitive parameter, and some further investigation is warranted. In addition, Table 9 and Table 10 demonstrate that CA, C, FA, MP and w have minimal effects on CC. This indicates that additional study is required to examine the input variables.

## Figures and Tables

**Figure 1 materials-15-05811-f001:**
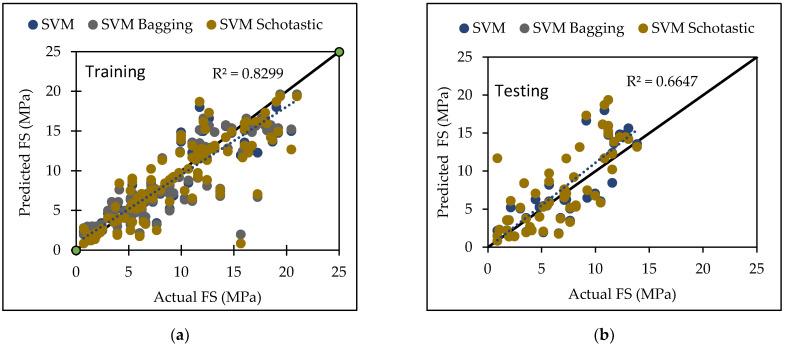
The scatter graph represents the observed and predicted FS values using SVM. (**a**) Training; (**b**) Testing.

**Figure 2 materials-15-05811-f002:**
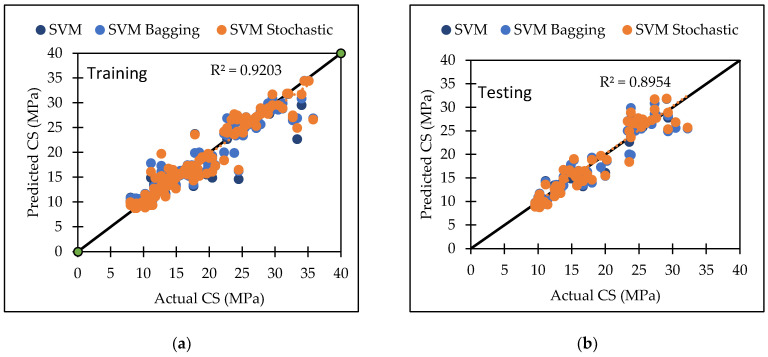
The scatter graph represents the observed and predicted CS values using SVM. (**a**) Training; (**b**) Testing.

**Figure 3 materials-15-05811-f003:**
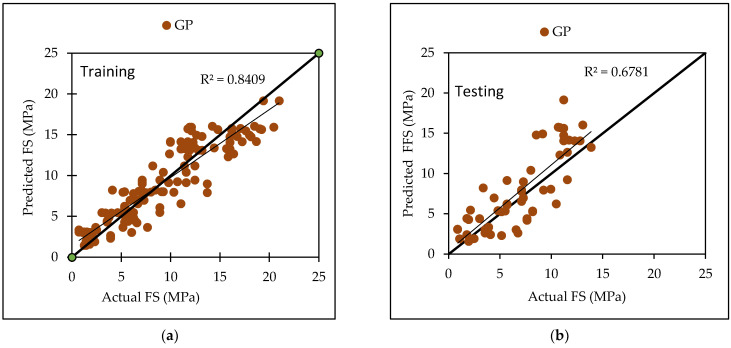
The scatter graph represents the observed and predicted FS values using GP. (**a**) Training; (**b**) Testing.

**Figure 4 materials-15-05811-f004:**
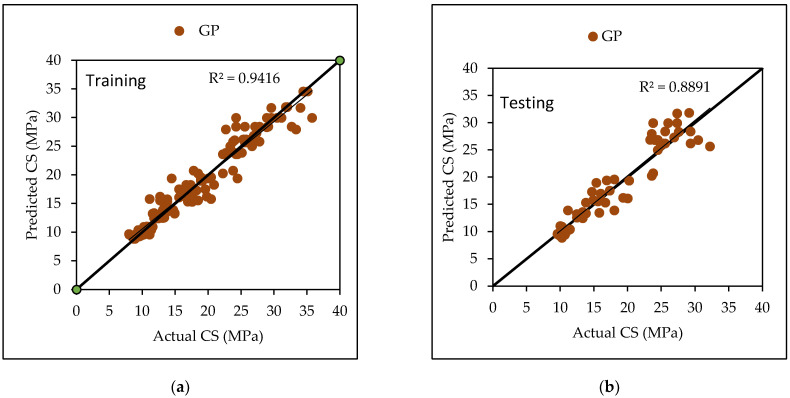
The scatter graph represents the observed and predicted CS values using GP. (**a**) Training; (**b**) Testing.

**Figure 5 materials-15-05811-f005:**
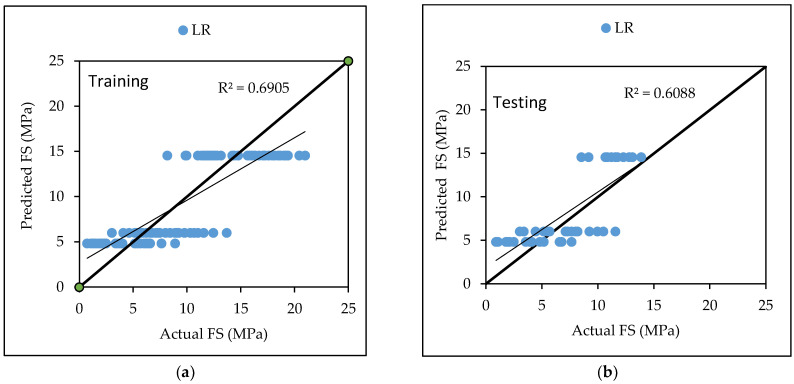
The scatter graph represents the observed and predicted FS values using LR. (**a**) Training; (**b**) Testing.

**Figure 6 materials-15-05811-f006:**
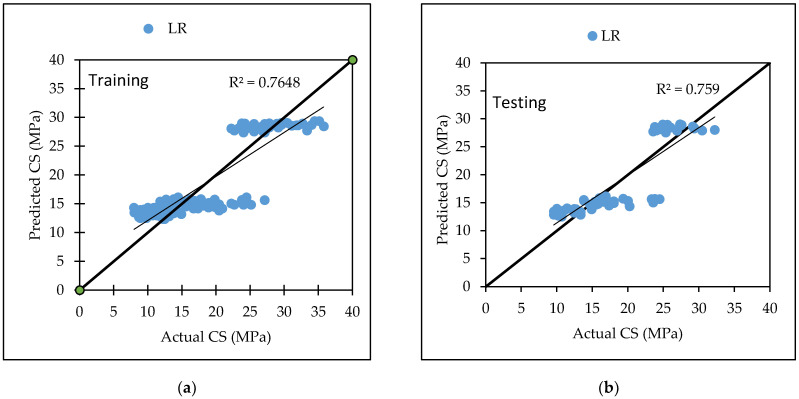
The scatter graph represents the observed and predicted CS values using LR. (**a**) Training; (**b**) Testing.

**Figure 7 materials-15-05811-f007:**
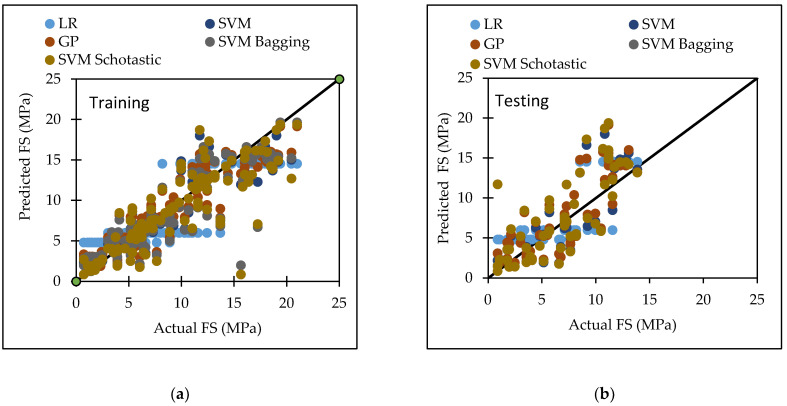
The scatter graph represents the observed and predicted FS values using SVM, GP, and LR. (**a**) Training; (**b**) Testing.

**Figure 8 materials-15-05811-f008:**
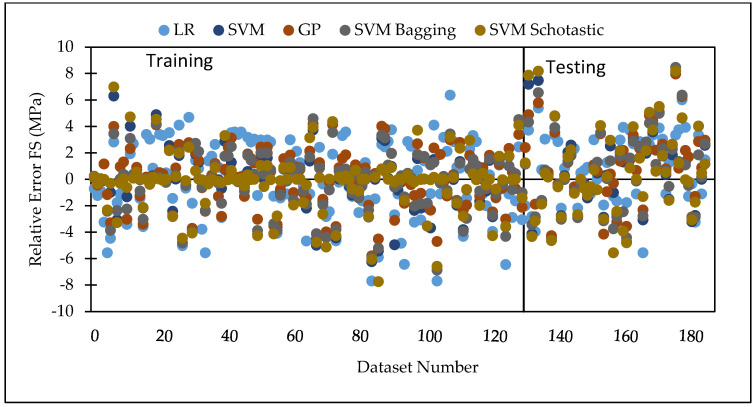
Error comparison between SVM, GP, and LR, for FS.

**Figure 9 materials-15-05811-f009:**
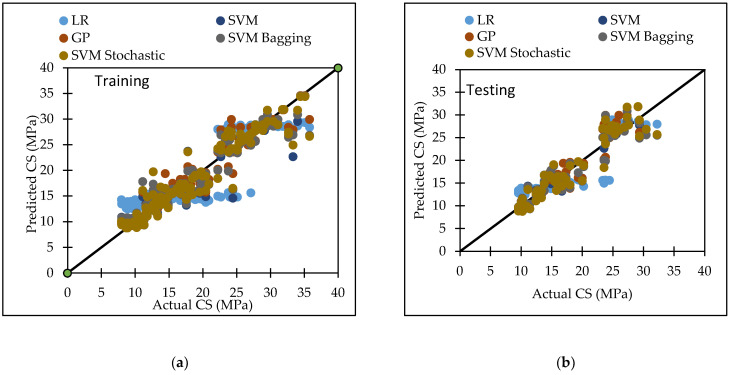
The scatter graph represents the observed and predicted CS values using SVM, GP, and LR. (**a**) Training; (**b**) Testing.

**Figure 10 materials-15-05811-f010:**
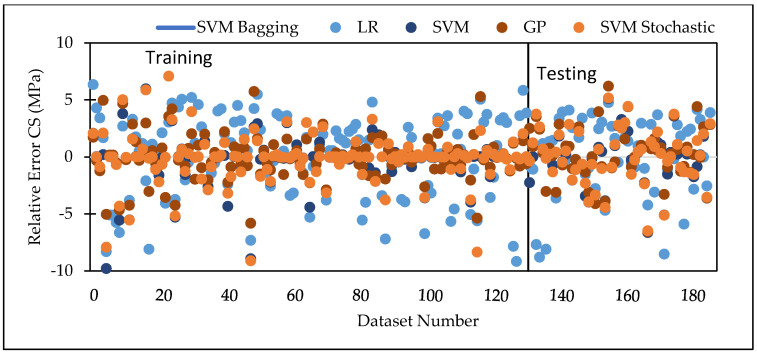
Error comparison between SVM, GP, and LR for CS.

**Figure 11 materials-15-05811-f011:**
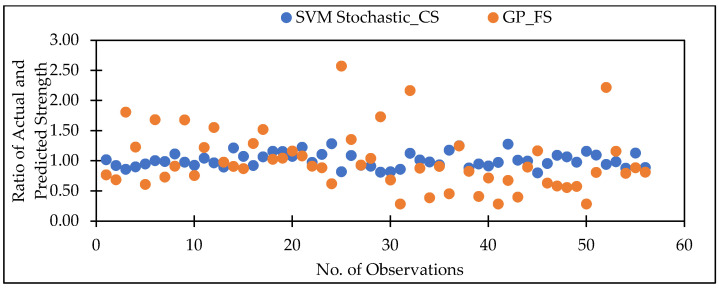
Scatter plot between No. of observation and ratio of actual and predicted strength for testing dataset.

**Figure 12 materials-15-05811-f012:**
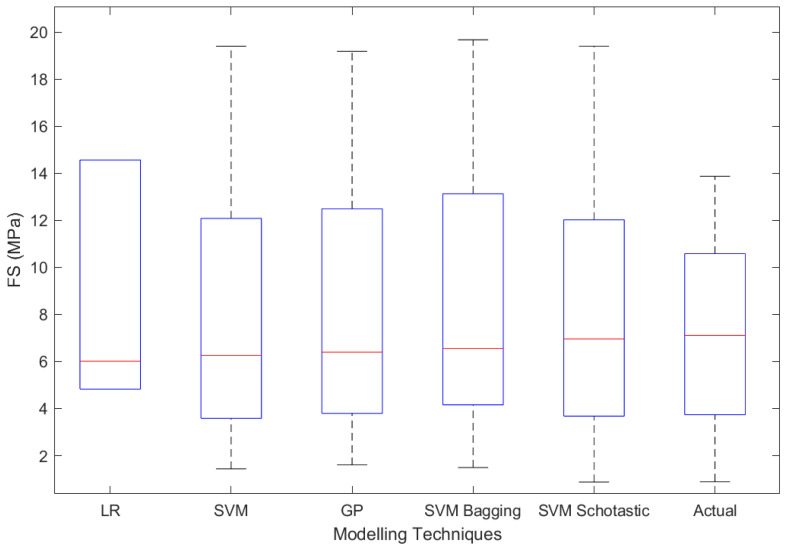
Box plot between modelling techniques and FS (testing).

**Figure 13 materials-15-05811-f013:**
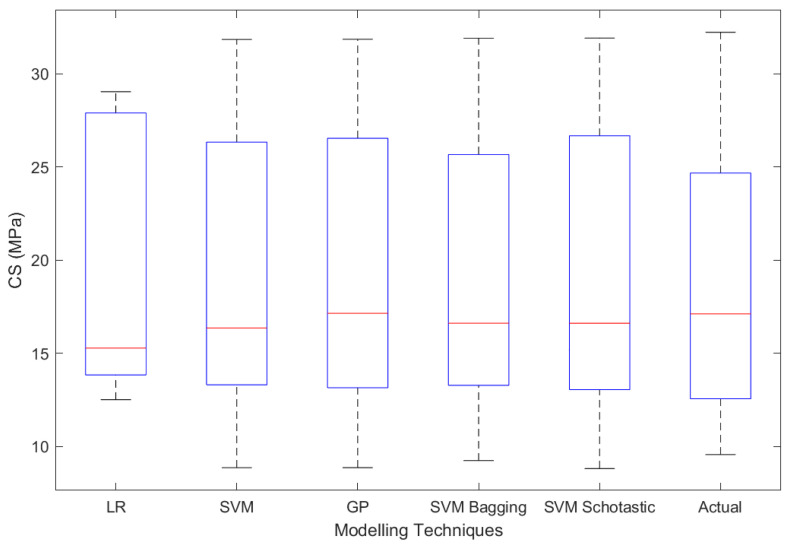
Box plot between modelling techniques and CS (testing).

**Table 1 materials-15-05811-t001:** Physical characteristics of aggregates.

Test	Unit	Value	Permissible Limit	Standard
Impact value test of coarse aggregate	%	6.74	<10	[52]
Crushing test of coarse aggregate	%	23.67	>45	[52]
SG (CA)	gm/cm^3^	2.61	-	[53]
Apparent SG (CA)	gm/cm^3^	2.82	-	
WA(CA)	%	2.82	-	
SG (FA)	gm/cm^3^	2.47	-	[53]
Apparent SG (FA)	gm/cm^3^	2.51	-	
WA (FA)	%	0.6	-	

**Table 2 materials-15-05811-t002:** Mix design of concrete.

Sr. No.	Mix-ID	Materials, kg/m^3^				
		C	FA	CA	w	MP
1	Conventional Mix	395.74	600.15	1103.09	217.65	0
2	MP 5(C0:S100)	395.74	570.14	1103.09	217.65	30.00
3	MP 5(C25:S75)	390.79	577.65	1103.09	214.93	27.44
4	MP 5(C50:S50)	385.85	585.15	1103.09	212.21	24.89
5	MP 5(C75:S25)	380.89	592.64	1103.09	209.49	22.34
6	MP 5(C100:S0)	375.95	600.15	1103.09	206.77	19.78
7	MP 10(C0:S100)	395.74	540.13	1103.09	217.65	60.01
8	MP 10(C25:S75)	385.85	555.13	1103.09	212.21	54.90
9	MP 10(C50:S50)	375.95	570.14	1103.09	206.77	49.75
10	MP 10(C75:S25)	366.06	585.15	1103.09	201.33	44.68
11	MP 10 (C100:S0)	356.16	600.15	1103.09	195.89	39.57
12	MP 15(C0:S100)	395.74	510.12	1103.09	217.65	90.02
13	MP 15(C25:S75)	380.89	532.63	1103.09	209.49	82.35
14	MP 15(C50:S50)	366.06	555.13	1103.09	201.33	74.69
15	MP 15(C75:S25)	351.21	577.65	1103.09	193.16	67.02
16	MP 15(C100:S0)	336.37	600.15	1103.09	185.00	59.36
17	MP 20(C0:S100)	395.74	480.12	1103.09	217.65	120.03
18	MP 20(C25:S75)	375.95	510.12	1103.09	206.77	109.80
19	MP 20(C50:S50)	356.16	540.13	1103.09	195.89	99.58
20	MP 20(C75:S25)	336.37	570.14	1103.09	185.00	89.43
21	MP 20(C100:S0)	316.59	600.15	1103.09	174.12	79.14

**Table 3 materials-15-05811-t003:** Characteristics of datasets for CS.

Dataset	Statistics	Minimum	Maximum	Mean	Standard Deviation	Kurtosis	Skewness
Total Dataset (189 observations)	C	316.59	395.74	372.18	22.28	−0.06	−0.89
	FA	480.12	600.15	564.43	33.79	−0.06	−0.89
	CA	1103.09	1103.09	1103.09	0.00	−2.02	1.01
	w	174.13	217.66	204.70	12.25	−0.06	−0.89
	MP	0.00	120.03	59.28	31.82	−0.88	0.10
	CD	7.00	180.00	71.67	77.29	−1.51	0.67
	CS	7.96	35.78	18.92	7.49	−1.1	0.34
Training Dataset (133 Observations)	C	316.59	395.74	372.08	23.51	−0.03	−0.95
	FA	480.12	600.15	563.83	35.96	−0.14	−0.91
	CA	1103.09	1103.09	1103.09	0.00	−2.03	1.01
	w	174.13	217.66	204.65	12.93	−0.03	−0.95
	MP	0.00	120.03	59.98	33.17	−0.88	0.08
	CD	7.00	180.00	72.32	77.76	−1.54	0.65
	CS	7.96	35.78	19.00	7.78	−1.09	0.37
Testing Dataset (56 Observations)	C	336.38	395.74	372.42	19.23	−0.74	−0.58
	FA	510.13	600.15	565.86	28.21	−0.59	−0.62
	CA	1103.09	1103.09	1103.09	0.00	−2.08	1.03
	w	185.01	217.66	204.83	10.58	−0.75	−0.58
	MP	0.00	109.80	57.62	28.58	−1.03	0.15
	CD	7.00	180.00	70.11	76.81	−1.44	0.74
	FS	0.89	13.87	6.91	3.7	−1.11	0
	CS	9.56	32.22	18.72	6.81	−1.32	0.24

**Table 4 materials-15-05811-t004:** Characteristics of datasets for FS.

Dataset	Statistics	Minimum	Maximum	Mean	Standard Deviation	Kurtosis	Skewness
Total Dataset (188 observations)	C	316.59	395.74	372.06	22.27	−0.07	−0.88
	FA	480.12	600.15	564.40	33.88	−0.08	−0.88
	CA	1103.09	1103.09	1103.09	0.00	−2.02	1.01
	w	174.13	217.66	204.63	12.25	−0.07	−0.88
	MP	0.00	120.03	59.43	31.83	−0.88	0.09
	CD	7.00	180.00	72.01	77.35	−1.52	0.67
	FS	0.71	20.98	8.19	4.98	−0.54	0.46
Training Dataset (132 Observations)	C	316.59	395.74	370.44	23.43	−0.21	316.59
	FA	480.12	600.15	563.95	34.47	−0.12	480.12
	CA	1103.09	1103.09	1103.09	0.00	−2.03	1103.09
	w	174.13	217.66	203.74	12.89	−0.21	174.13
	MP	0.00	120.03	61.50	31.56	−0.75	0.00
	CD	7.00	180.00	76.43	79.04	−1.69	7.00
	FS	0.71	20.98	8.73	5.35	−0.84	0.71
Testing Dataset (56 Observations)	C	336.38	395.74	375.86	18.92	−0.28	−0.89
	FA	480.12	600.15	565.45	32.73	0.12	−0.97
	CA	1103.09	1103.09	1103.09	0.00	−2.08	1.03
	w	185.01	217.66	206.73	10.41	−0.28	−0.89
	MP	0.00	120.03	54.57	32.23	−0.91	0.42
	CD	7.00	180.00	61.59	72.83	−0.90	1.02
	FS	0.89	13.87	6.91	3.70	−1.11	0.00

**Table 5 materials-15-05811-t005:** Use defined parameters.

Model Used	User Defined Parameters	
	CS	FS
**LR**	S = 0	S = 0
**SVM**	C = 5, RBF(Ɣ = 10)	C = 1.5, PUK (O = 4, S = 1)
**SVM Bagging**	C = 10, RBF(Ɣ = 11)	C = 1.5, PUK (O = 5, S = 1)
**SVM Stochastic**	C = 1.5, PUK (O = 1, S = 1)	C = 1.5, PUK (O = 5, S = 1)
**GP**	L = 0.1 and O = 0.1 and S = 1.0.	L = 0.01, O = 3 and S = 3

**Table 6 materials-15-05811-t006:** Performances of SVM, GP, and LR for FS.

Approaches	CC	MAE	RMSE	RAE	RRSE
**Training**					
GP	0.9170	1.7012	2.1276	37.45%	39.90%
LR	0.8309	2.4992	2.9666	55.01%	55.64%
SVM	0.9110	1.5123	2.2287	33.29%	41.80%
SVM-Bagging	0.9149	1.596	2.1456	35.62%	40.57%
SVM-Schotastic	0.9115	1.3424	2.2013	29.96%	41.62%
**Testing**					
GP	0.8235	2.2808	2.8527	66.00%	69.64%
LR	0.7802	2.3822	2.7717	68.94%	67.66%
SVM	0.8153	2.3456	3.0367	67.88%	74.13%
SVM-Bagging	0.8088	2.3148	2.9142	65.92%	70.05%
SVM-Schotastic	0.7815	2.4504	3.2176	69.78%	77.34%

**Table 7 materials-15-05811-t007:** Performances of SVM, GP, and LR for CS.

Approaches	CC	MAE	RMSE	RAE	RRSE
**Training**					
GP	0.9704	1.2800	1.8734	18.82%	24.17%
LR	0.8745	3.0323	3.7593	44.58%	48.50%
SVM	0.9545	1.2472	2.3289	18.34%	30.04%
SVM-Bagging	0.9636	1.4074	2.113	20.69%	27.25%
SVM-Schotastic	0.9593	1.2541	2.1936	18.43%	28.29%
**Testing**					
GP	0.9429	1.8728	2.4377	30.80%	36.11%
LR	0.8712	2.6909	3.3921	44.26%	50.24%
SVM	0.9450	1.7305	2.2908	28.46%	33.93%
SVM-Bagging	0.9404	1.7678	2.3477	29.07%	34.77%
SVM-Schotastic	0.9462	1.8104	2.343	29.77%	34.70%

**Table 8 materials-15-05811-t008:** Statistics of actual and predicted CS and FS values of testing dataset using various soft computing techniques.

Investigated Property	Statistic	GP	LR	SVM	SVM Bagging	SVM Schotastic	Actual
Flexural Strength	Minimum	1.60	4.83	1.43	1.49	0.88	0.89
	Maximum	19.18	14.56	19.40	19.67	19.40	13.87
	1st Quartile	4.01	4.83	3.59	4.19	3.72	3.82
	Mean	7.81	7.90	7.66	7.92	7.96	6.91
	3rd Quartile	12.40	14.56	11.90	13.09	11.87	10.54
	IQR	8.40	9.74	8.31	8.90	8.15	6.71
Compressive Strength	Minimum	8.85	12.50	8.85	9.23	8.80	9.56
	Maximum	31.84	29.03	31.83	31.90	31.91	32.22
	1st Quartile	13.19	13.84	13.33	13.36	13.19	12.61
	Mean	19.00	18.84	18.72	18.68	18.83	18.71
	3rd Quartile	26.37	27.89	26.06	25.66	26.66	24.55
	IQR	13.17	14.05	12.73	12.29	13.46	11.94

**Table 9 materials-15-05811-t009:** Sensitivity analysis results using a GP-based model for FS.

Removed Parameter	GP Based Model		
	CC	MAE	RMSE
-	0.8235	2.2808	2.8527
CD	0.0900	3.5782	4.2731
CA	0.8235	2.2808	2.8527
C	0.8251	2.2193	2.7812
w	0.8251	2.2193	2.7811
MP	0.8263	2.112	2.7555
FA	0.8335	2.1823	2.7226

**Table 10 materials-15-05811-t010:** Sensitivity analysis results using SVM-based model for CS.

Removed Parameter	SVM-Schotastic Based Model		
	CC	MAE	RMSE
-	0.9462	1.8104	2.3430
CD	0.0872	5.8869	7.2242
C	0.9440	1.8569	2.4131
W	0.9443	1.8504	2.4066
FA	0.9459	1.7980	2.3270
MP	0.9461	1.7495	2.3227
CA	0.9462	1.8104	2.3430

## Data Availability

Not applicable.

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
