# Peer review of "Machine Learning Techniques for Evaluating Concrete Strength with Waste Marble Powder"

_materials, 2022, doi:10.3390/ma15175811_

Round 1

Reviewer 1 Report

Paper deals with important problem. The authors considered a prediction task for the compressive and flexural strengths of the concrete mix by using waste marble powder based on the experimental data that was acquired from the laboratory tests.

Paper has scientific novelty and great practical value.

It has a logical structure all necessary sections. The paper is technically sound. The experimental section is good.

The proposed approach is logical, results are clear.

Suggestions:

1.         It would be good to add point-by-point the main contributions at the end of the Introduction section

2.         It would be good to add the remainder of this paper

3.         The related works section should be extended using hybrid SVM-based approaches for solving the stated task. The authors can use some of these papers: DOI: 10.1016/j.procs.2021.07.029,  DOI: 10.3390/app12105238 among others.

4.         The authors should provide a link to open access repository with the dataset used for modelling.

5.         The authors should add all optimal parameters for all investigated methods

6.         It would be good to see also R2 values for all investigated methods.

7.         The conclusion section should be extended using: 1) numerical results obtained in the paper; 2) limitations of the proposed approach; 3) prospects for future research.

8.         A lot of references are outdated. Please fix it using 3-5 years old papers in high-impact journals.

Author Response

Response to reviewer -1

Reviewer 2 Report

Before the article is accepted for publication, please verify the information below:

- Include in the abstract the main conclusions of the research. Note that in its current state the abstract does not present the main findings obtained in the article.

- Review the grammar errors present in the introduction. Some sentences are difficult to understand. Review the text formatting in the introduction. Some texts are misaligned.

- A more in-depth approach to the effect of Marble waste on cement materials was expected. The authors approach some works, but the presentation of information is scarce. I suggest including other works that evaluated the incorporation of this material, such as: 10.1007/s10163-019-00878-6

- There are many Marble waste works in cement materials. That is why it is important to include at the end of the introduction what are the main novelties of the research. What is being investigated again that makes the article useful for publication?

- Section 3.2 of the article is difficult to understand. The authors focus on explaining the methodology of the modeling part of the research, but fail to present the methodologies of the experimental part. Add standards used, mixing techniques, sample molding techniques, curing conditions, quantity of specimens used in the research.

- It is extremely important to include information on the granulometry of the materials used in the research, especially aggregates and Marble waste. Please include this information in the manuscript.

- Standardize the figures present in your manuscript. This is important to compare the results with each other.

- Based on the results obtained from properties (mainly Flexural Strength and Compressive Strength), what are the applications of the material studied in your research?

- It is important to compare the results obtained with similar surveys. There are no comparisons of results with authors or other similar research. This is not acceptable in a scientific text. Please include a discussion of this in your revised manuscript.

- Organize the conclusion into topics (from 3 to 5) that present the main conclusions found in your manuscript. In the current format the conclusion is not adequate. Also review your conclusion based on previous comments.

Author Response

Response to reviewer -2

Round 2

Reviewer 1 Report

The authors took into account all comments of the reviewer and made appropriate corrections to the manuscript.

Reviewer 2 Report

The authors responded to comments on the previous version. Therefore, I recommend the article for publication in its current format.